# The Role of Self-Compassion in the Job Demands-Resources Model, an Explorative Study among Crisis Line Volunteers

**DOI:** 10.3390/ijerph18189651

**Published:** 2021-09-13

**Authors:** Renate Willems, Constance Drossaert, Peter ten Klooster, Harald Miedema, Ernst Bohlmeijer

**Affiliations:** 1Research Center Innovations in Care, Rotterdam University of Applied Science, 3015 EK Rotterdam, The Netherlands; h.s.miedema@hr.nl; 2Department of Psychology, Health and Technology, University of Twente, 7522 NB Enschede, The Netherlands; c.h.c.drossaert@utwente.nl (C.D.); p.m.tenklooster@utwente.nl (P.t.K.); e.t.bohlmeijer@utwente.nl (E.B.)

**Keywords:** job demands-resources model, self-compassion, personal resources, volunteering, crisis line

## Abstract

The job demands-resources (JD-R) model has hardly been studied in volunteer organizations and there is a scarcity of studies evaluating self-compassion as a personal resource within the JD-R model. The present study addresses these gaps in current knowledge, first by examining the applicability of the JD-R model in a crisis line volunteer organization. Second, self-compassion is examined, both in terms of its moderating role on the exhaustion process as well as its role on the motivation process. Structural equation modelling was used for the analyses. The influence on the organizational outcome ‘compassion towards others’ was examined using a multiple regression analysis. The results showed that the JD-R model has an acceptable fit on this sample and supports the central assumption that exhaustion and motivation are two independent but related processes. This study provides evidence that self-compassion is a valuable addition to the JD-R model, as it has an indirect effect on both processes, and increases the explained variance in compassion towards others by 7% through the exhaustion process and by 3% through the motivational process. These findings point to the importance of focusing on self-compassion in training and supervision in volunteer organizations.

## 1. Introduction

During the past decades, there has been increasing attention for the well-being of employees in all kinds of occupations. The job demands-resources model (JD-R model) is a well-studied model that provides insight into the factors potentially influencing the well-being of employees, providing direction for interventions to improve their well-being [1,2,3]. In this model, working conditions are divided into *job demands* (physical, psychological, social, or organizational aspects of the work that require effort and/or skills) and *job resources* (those physical, psychological, social, or organizational aspects of the work that help to achieve goals and reduce work-related stress). Both job characteristics may influence the level of employees’ distress and engagement. The JD-R model consists of two underlying psychological processes: the exhaustion and motivational processes. The exhaustion process predicts that high work demands in the absence of job resources cause distress. The motivational process predicts that the presence of job resources will contribute to higher engagement and productivity [1,2,3]. Together, these processes predict organizational outcomes, such as commitment and turnover intentions [1,2,3].

Previous research has found ample support for this model, including support for the relationship between job demands and distress, and for the relationship between job resources and engagement [4,5,6]. In addition, it was shown that job resources act as a buffer in the relationship between job demands and distress [7,8]. Evidence was also found for the impact of the exhaustion and motivational processes on organizational outcomes. A recent study on social workers’ commitment and turnover intention, using the dual process in the JD-R model, showed that social workers’ task demands (i.e., workload) influenced their intention to leave the organization, while their jobs’ resources (i.e., training) predicted their commitment to the organization [9].

The JD-R model has been studied mainly among paid employees. Less is known about the applicability of the JD-R model among volunteers. Many organizations depend on volunteers who provide valuable health care [10] and make society more healthy, just, and sustainable [11]. Exploring the applicability of the JD-R model in a volunteer setting is thus vital and the aim of this study. We were specifically interested in crisis line volunteers who offer a listening ear 24/7 to anyone who cannot or does not want to make use of professional help [12,13]. There are a number of potential job demands related to this work, including the suffering of other people, such as loneliness, suicidality, and mental and physical suffering [13]. In addition, the inappropriate behavior of callers who make sexual, abusive, and manipulative calls can contribute to job demands [14]. These stressors can result in reduced mental wellbeing [15]. Potential job resources, on the other hand, include training, supervision, and co-worker support (job resources), which help the volunteers cope with their job demands [15]. Crisis line volunteers can experience a high degree of distress [15,16]. In addition, crisis line volunteers are highly motivated and engaged in this volunteer work [15,17]. However, limited scientific research is known in which the association between specific job demands and job resources with distress and engagement is proven for crisis line volunteers. The crisis line service acts from the presence approach [18], meaning that the crisis line volunteer relates to the other with attention and dedication, develops understanding for the meaning of the suffering of the other, and has the intention to alleviate this suffering. Being present also means not abandoning the other person, neither emotionally nor relationally, but rather staying involved with the other person [19]. This approach is very similar to compassion. Gilbert [20] describes compassion as “A deep awareness of the suffering of another coupled with the desire to relieve it.” Therefore ‘compassion to others’ seems to be the major organizational outcome of crisis line services. The JD-R model can be useful to study factors that impact the distress and engagement of crisis line volunteers and their relationship with compassion towards others as a vital organizational outcome.

In recent years, an increasing number of researchers have studied personal resources as a new addition to the JD-R model, because human behavior results from an interaction between the person and the work context [3]. Personal resources are characteristics of the person that are related to their resilience and ability to influence the work environment, such as self-confidence, self-efficacy, and optimism. They can enable a person to achieve work goals and encourage personal growth [3]. There are various ways in which personal resources can be added to the JD-R model. First, personal resources can have a direct, positive effect on engagement. A higher degree of engagement may subsequently result in an increased use of job resources such as promoting the supportive interaction between colleagues [21]. Second, personal resources can buffer the exhaustion process. For example, self-efficacy has been shown to be a significant moderator in the relationship between job demands and burnout [22]. Third, evidence has also been found for personal resources as mediators in the motivational process. For example, the relationship between job resources and engagement was shown to be mediated by the personal resource ‘emotion regulation’ among university teachers [23].

A potentially relevant personal resource is self-compassion. There is a growing academic interest in the relevance of (self-)compassion in the context of organizations. Lilius et al. [24], for example, showed that employees who experienced compassionate care when they were suffering, experienced greater connection and engagement with their work. Furthermore, self-compassion has also been shown to promote work performance and prosocial behavior, and to reduce negative work experiences, such as emotional exhaustion and turnover intention [25]. Various models have been developed to describe self-compassion. Neff [26] described self-compassion as: “being touched by and open to one’s own suffering, not avoiding or disconnecting from it, generating the desire to alleviate one’s suffering and to heal oneself with kindness”. Another influential theory on (self-)compassion was created by Gilbert [20]. He defines (self-) compassion as: “A deep awareness of the suffering of another (or oneself) coupled with the wish to relieve it”. Strauss et al. [27] conducted a comprehensive review to bring together the various definitions of (self-) compassion. Based on this review, they defined self-compassion as a cognitive, affective, and behavioral skill, consisting of the following five elements: recognizing suffering; understanding the universality of suffering; feeling empathy for one’s own suffering; tolerating uncomfortable feelings; and acting, or being motivated to act, to alleviate suffering. Research shows that self-compassion can enhance resilience [28], engagement, and mental well-being, and reduce distress [29,30] and burnout [31,32]. In addition, it has shown to be a predictor of increased mental wellbeing [33,34] and engagement [35,36]. Since self-compassion is an important resource for coping with negative emotions and cognitions resulting from stressors and adversity, it may buffer the relationship between job demands and distress. Self-compassion may also mediate the motivational process. For example, job resources, such as training, supervision, and co-worker support could enhance self-compassion, in turn leading to greater engagement. Both lower distress and larger engagement could positively impact desired organizational outcomes.

However, despite the increasing scholarly attention for the concept of self-compassion, and its potential contribution to the JD-R model, three studies have examined the role of self-compassion as a personal resource within the JD-R model. Anjum et al. [37] found that self-compassion has a moderating effect on the relationship between job demands (being bullied and excluded at work) and exhaustion among employees of various service sector organizations. Monaghan et al. [38] did not find evidence for the moderating effect of self-compassion on the relationship between job demands and exhaustion among animal care professionals; however, there was a strong association between burnout and self-compassion (α = −0.50). In addition, in a study among physician assistants it was found that self-compassion increased engagement and reduced the risk of burnout [39]. 

To summarize, the JD-R model has hardly been studied in volunteer organizations and there is a scarcity of studies comprehensively evaluating self-compassion as a personal resource within the JD-R model, despite the growing evidence on its impact on mental functioning. The present study addresses these gaps in current knowledge. 

We hypothesize that:the original JD-R model is applicable to crisis line volunteers;self-compassion acts significantly as a buffer in the exhaustion process;self-compassion acts significantly as a mediator in the motivational process;self-compassion significantly increases the explained variance of compassion towards others, through the exhaustion and motivational processes.

## 2. Materials and Methods

### 2.1. Aims and Design

A survey study was performed to explore the JD-R model and the role of self-compassion in a crisis line volunteer organization. The study was approved by the ethical committee of the Faculty of Behavioral and Management Studies (BMS) of the University of Twente (approval number: 190943).

### 2.2. Procedure

This study was conducted among volunteers of the ‘Listening Line’, a Dutch crisis line organization that offers an active, non-judgmental listening service to anyone who cannot or does not want to make use of professional care [40]. All crisis line volunteers (*N* = 1405) received a link to the online questionnaire by e-mail from their management, explaining the purpose and content of the study. After giving their consent, volunteers could continue to complete the anonymous questionnaire.

### 2.3. Measurements

*Job demands:* Job demands were measured by two questionnaires measuring: work-related demands and emotional strain. *Work-related demands* were measured using a 16-item self-developed questionnaire [41]. Each demand described a potentially distressing situation that a crisis line volunteer may encounter, and two questions were asked. The first question related to the occurrence of the situation (‘How often does this situation occur?’) with answering options ranging from ‘never’ (0) to ‘very often’ (4). The second question related to the degree of stress that this situation causes (‘How stressful is this situation for you?’) with answering options ranging from ‘not at all stressful’ (1) to ‘very stressful’ (5). The impact of each stressor was calculated by multiplying the frequency of occurrence with the degree of stress produced by the stressor. This questionnaire was created based upon the results of a systematic review on the work-related demands of crisis line volunteers and a qualitative study among crisis line volunteers about the emotional impact, challenges, and resources available to them for coping with the challenges of volunteering [15,42]. As a pilot test, the questions were presented to ten crisis line volunteers, who found the items recognizable and clear. Upon their reactions, only a few minor textual changes were made. *Emotional strain* was measured with the subscale ‘emotional strain’ of the Experience and Assessment of Work questionnaire [43], a Dutch questionnaire that measures work perception in many areas. An example of one of the items is: “In your work, are you confronted with issues that affect you personally?”. This subscale consists of seven items, scored on a four-point Likert scale, from 1 (never) to 4 (always).

*Job resources:* Job resources were measured by combining self-developed questions with questions from an existing questionnaire. The questions covered three themes: co-worker support, training, and supervision. To measure co-worker support, the subscale ‘relation with colleagues’ of the Experience and Assessment of Work questionnaire [43] was used. An example of one of the items is: “Can you ask your colleagues for help when needed?”. This subscale consists of 9 items, scored on a four-point Likert scale, from 1 (never) to 4 (always). The questions about training (5 items) and supervision (4 items) were self-developed questions, scored on a Likert scale from 1 (totally agree) to 5 (totally disagree). The items in the questionnaire were also, like the job demands, created on the basis of the results of a systematic review and a qualitative study of crisis line volunteers [15,42], pilot tested among ten crisis line volunteers, which led to only a few minor changes in the wording of the questions.

*Distress*: Distress was measured with the subscale distress of the validated Four Dimensional Complaint List (4-DCL) [44]. The subscale distress contains 16 items, scored on a five-point Likert scale, ranging from 1 (never) to 5 (always). The occurrence of the presence of distress was determined by reducing the five answer categories of the Likert scale to three answer categories (never = 0, sometimes = 1, regularly or more often = 2), and subsequently summing the items to a total score, ranging from 0 to 32. Based upon these scores, participants were categorized in low (0–10), moderately increased (11–20), or strongly increased distress (21–32), as outlined in the 4DSQ manual [44].

*Engagement*: Engagement was measured with the validated Utrecht Work Engagement Scale short version (UWES-9) [45]. This scale consists of 9 items that can be scored on a seven-point Likert scale ranging from 1 (never) to 7 (always). The mean total score was categorized as very low (<1.77), low (1.78–2.88), moderate (2.89–4.66), high (4.67–5.50), and very high (>5.51) [46].

*Organizational outcome:* Compassion towards others was measured using the ‘compassion towards others’ subscale of the validated Compassionate Engagement and Action Scales (TCEAS) [47]. Because of the anonymity of the callers and chatters, it is not possible to have them rate the level of compassion given by volunteers. Therefore, it was decided to use this self-assessment questionnaire. This subscale consists of 13 items that can be scored on a 10-point Likert scale ranging from 1 (never) to 10 (always). Following the manual, three reversed items were removed before constructing the scale.

*Self-compassion*: Self-compassion was measured with the Dutch version of the Sussex-Oxford Compassion for the Self Scale (SOCS-S) [48], a 20-item questionnaire, scored on a 5-point Likert scale, ranging from 1 (not at all true) to 5 (always true). The scale consists of five subscales: ‘Recognizing suffering’, ‘Understanding the universality of suffering’, ‘Feeling for one’s own suffering’, ‘Tolerating uncomfortable feelings’, and ‘Acting or being motivated to act to alleviate suffering’.

### 2.4. Statistical Analyses

Statistical analyses were performed using IBM SPSS Statistics (version 26, IBM Corp, Armonk, NY, USA) and AMOS (version 25, IBM Corp, Armonk, NY, USA).

Descriptive statistics were used to analyze background and work-related variables, characteristics of all variables, and Pearson correlations between the variables.

To examine the extent to which the JD-R model is applicable to crisis line volunteers, structural equation modelling (SEM) with maximum likelihood estimation, was used. Total model fit was tested using the root mean square error of approximation (RMSEA), the standardized root means square residual (SRMR), the comparative fit index (CFI), and the goodness of fit index (GFI). The RMSEA and the SRMR should preferably be <0.08, the CFI should preferably be >0.90, and the GFI should preferably be >0.95 [49] for adequate fit. The conventional chi-square test is almost always significant in large samples and therefore likely to overstate the lack of fit [49]. 

SEM analyses were performed to examine whether self-compassion buffers the exhaustion process, and whether self-compassion mediates the motivational process. Moderation and mediation analyses were performed separately for the specific part of the JD-R model that the moderation or the mediation refers to, namely the relationship between job demands and distress, and the relationship between job resources and engagement. Each variable had only one indicator, namely its standardized score. A significant interaction effect exists if the 95% bootstrapped confidence interval (2000 bootstraps) of the interaction coefficient does not contain zero. 

A hierarchical stepwise multiple regression was used to explore whether self-compassion is a unique predictor of compassion towards others, in addition to demographics and work-related variables (step 1), to job demands and job resources (step 2), and to distress and engagement (step 3).

## 3. Results

### 3.1. Descriptives

The total number of respondents was 543 (response rate 39%). Table 1 provides a summary of the demographic and work-related variables. The majority of the participants were female and over 50 years old. Most of them had no professional training in health care, worked 4–6 h a week at the crisis line services, and had 1–3 years of experience in working as a crisis line volunteer.

Table 2 gives an overview of the descriptive statistics of all variables including a correlation matrix. The Cronbach’s alphas of almost all variables were good (α > 0.70), only that of emotional strain was low but acceptable (α = 0.61). The bivariate correlations between the different variables met the expected directions, except for the correlation between distress and compassion towards others, which was expected to be negative. 

The Cronbach’s alphas in the current study of the subscales of self-compassion (not in the table) were: ‘Recognizing suffering’ α = 0.81, ‘Understanding the universality of suffering’ α = 0.85, ‘Feeling for the person suffering’ α = 0.85, ‘Tolerating uncomfortable feelings’ α = 0.84, and the subscale ‘Acting or being motivated to act to alleviate suffering’ α = 0.87.

According to the classification of Terluin et al. (2014) a total of 474 respondents (82%) scored low, 79 (15%) scored moderate, and 17 (3%) scored high on distress. According to the classification of Schaufeli and Bakker (2003) a total of 326 respondents (63%) scored (very) high, 184 (36%) scored moderate, and 6 (1%) scored (very) low on engagement.

### 3.2. The Job Demands-Resources Model and Crisis Line Volunteers

For testing whether the JD-R model can be applied to crisis lines, an SEM analysis was used. The specified JD-R model (Figure 1) demonstrated acceptable fit indices to the data: Chi-square (df = 16) = 84.9, *p* < 0.001; GFI = 0.96; CFI = 0.90; RMSEA = 0.09; SRMR = 0.06. Figure 1 shows the standardized relations between the variables of the JD-R model.

The relationship between job demands and distress, and the relationship between job resources and engagement are positive and moderate in strength. The results show that there is a significant positive relationship between engagement and compassion towards others, but no significant relationship between distress and compassion towards others. No moderating effect of job resources (training, supervision, co-worker support) on the relationship between job demands (work-related demands and emotional strain) and distress was found. Furthermore, no evidence was found for the moderating role of job demands (work-related demands and emotional strain) on the relationship between job resources (training, supervision, and co-worker support) and engagement.

### 3.3. The Role of Self-Compassion on the Exhaustion and Motivational Processes

To examine the added value of self-compassion in the JD-R model, moderation analyses for the exhaustion process and mediation analyses for the motivational process were conducted.

#### 3.3.1. The Moderating Role of Self-Compassion on the Exhaustion Process

In order to test the moderating influence of self-compassion on the relationship between job demands and distress, SEM analyses were conducted in two separate models. First, a model included three exogenous variables (work-related demands, self-compassion, and their interaction), and one endogenous variable (distress). Second, the same model was tested, but with emotional strain as an exogenous variable instead of work-related demands.

The first model, the moderation analysis of self-compassion on the relationship between work-related demands and distress, showed significant and negative interaction between self-compassion and work-related demands (*β* = −0.13, *p* < 0.01), indicating that self-compassion dampens the positive relationship between work-related demands and distress. Figure 2a illustrates this buffering effect of self-compassion in an interaction plot.

Since self-compassion has a moderating effect on the relationship between work-related demands and distress, moderation analyses were also performed with the separate subscales of self-compassion. Significant and negative moderation effects were found for recognizing suffering (*β* = −0.08, *p* < 0.05), feeling suffering (*β* = −0.13, *p* < 0.01), tolerating feelings (*β* = −0.12, *p* < 0.01), and action to alleviate suffering (*β* = −0.15, *p* < 0.001), indicating that these subscales dampen the positive relationship between work-related demands and distress (Figure 2b–e). No significant moderation effect for understanding the universality of suffering was found.

The second model, the moderation analysis of self-compassion on the relationship between emotional strain and distress shows a similar significant and negative interaction between self-compassion and work-related demands (*β* = −0.12, *p* < 0.01), indicating that self-compassion also dampens the positive relationship between emotional strain and distress (Figure 3a).

Similar to work-related demands, significant and negative interaction effects were found for recognizing suffering (*β* = −0.11, *p* < 0.05), feeling suffering (*β* = −0.09, *p* < 0.05), tolerating feelings (*β* = −0.13, *p* < 0.001), and action to alleviate suffering (*β* = −0.11, *p* < 0.01) and emotional strain, indicating that these aspects of self-compassion also dampen the positive relationship between emotional strain and distress (Figure 3b–e). Again, no significant moderation effect by understanding the universality of suffering was found.

#### 3.3.2. The Mediating Role of Self-Compassion on the Motivational Process

In order to test the mediating role of self-compassion in the relationship between job resources and engagement, SEM analyses were carried out in three separate models: first, a model with training as the exogenous variable, engagement as the endogenous variable, and self-compassion as the mediating variable; second, the same model, but with supervision as an exogenous variable; third, the same model, but with co-worker support as an exogenous variable. 

Table 3 shows the results of the mediation analyses for each model. Because self-compassion is a significant mediator in all three models, each subscale was separately tested as a mediator in each model. Self-compassion and all the subscales, except ‘acting/being motivated to act to alleviate suffering’ mediated the relationship between all job resources and engagement. ‘Acting or being motivated to act to alleviate suffering’ only mediated the relationship between training and engagement.

### 3.4. The Added Value of Self-Compassion in Predicting Compassion towards Others

A summary of the multiple regression analysis of the added value of self-compassion for explaining compassion towards others in the exhaustion process and the motivational process of the JDR model is shown in Table 4. 

Demographics and work-related variables did not significantly explain the variance of compassion towards others. When job demands were added to the model, the explained variance of compassion towards others increased significantly to 3%. The addition of distress did not significantly increase the total explained variance of compassion towards others. When self-compassion was added, the total explained variance of compassion towards others in the exhaustion process increased to 10% (*F_change_* = 39.8, *p* < 0.001).

The addition of job resources to demographics and work-related variables accounted for 9% of the total explained variance of compassion towards others. This increased to 14% when engagement was added. When self-compassion was added as well, the total explained variance of compassion towards others within the motivational process increased to 17% (*F_change_* = 15.5, *p* < 0.001).

## 4. Discussion

The aim of the current study was to examine the applicability of the JD-R model to a volunteer organization, and to explore the added value of self-compassion as a personal resource to the JD-R model.

### 4.1. The Applicability of the JD-R Model on Volunteers

The findings of this study demonstrated an acceptable fit of the JD-R model in a sample of 543 crisis line volunteers. The findings also support the central assumption of the JD-R model that there are two independent but related processes, exhaustion and motivation, that influence the organizational outcome [1,3]. We found evidence for the relationship between job demands and distress and job resources and engagement. This is in line with results from research conducted with paid workers [50,51,52] and studies with volunteers [53,54]. The relationship between engagement and ‘compassion towards others’ as an organizational outcome was found to be significant. This finding was expected, as altruism and the desire to help others are important motivations for crisis line volunteers [55,56,57]. In this study, no significant relationship was found between distress and the organizational outcome ‘compassion towards others’. This finding is against expectation as the literature describes that distress caused by exposure to the suffering of others can lead to compassion fatigue, i.e., ‘empathic strain and general exhaustion resulting from dealing with people in distress over time’ [58,59]. The low mean score on distress in this sample (*M* = 6.3, SD = 5.5, range = 0–32) may have suppressed a relationship between distress and compassion towards others. Another possible explanation lies in the choice of the outcome measure ‘compassion towards others’. Previous studies on the application of the JD-R model to volunteers, have focused on using ‘turnover intention’ [53,54] as an organizational outcome measure and found significant relationships with this outcome variable in the exhaustion process. Volunteers who experience distress and reduced job satisfaction may have the intention to stop volunteering, but may still be able to suppress their own feelings of stress in order to be compassionate towards others. Previous research has shown that there is a close correlation between stress and compassion towards others [47]. Further research could focus on personal outcome measures, such as depression or mental wellbeing, and other organizational outcome measures, such as turnover intention.

We found no moderating influence of job demands on the motivational process and job resources on the exhaustion process. A possible explanation is the combination of very low mean scores on job demands and distress and very high mean scores on job resources and engagement. It is important to note that crisis line volunteers at the Dutch Listening Line are carefully selected. In this selection process, attention is paid to the task requirements they are going to face [60]. Volunteers who find out beforehand that they cannot cope with the challenges of the job may decide in time not to volunteer.

It can be concluded that the JD-R model is applicable to volunteer organizations. Specifically, the motivational process was found to play an important role in explaining the organizational outcome measure ‘compassion towards others’.

### 4.2. Self-Compassion as an Added Value to the JD-R Model

The second aim of the current study was to explore the added value of self-compassion as a personal resource to the JD-R model.

We found that self-compassion buffers the relationship between job demands and distress. The crisis line volunteers in our sample also scored higher (M = 78.1, SD = 9.4) than a comparable group (M = 70.8, SD = 11.7) on self-compassion [48]. Self-compassion supports facing feelings of inadequacy and stressors with kindness and self-reassurance [61]. The awareness that suffering happens to everyone and the ability to see this suffering in the right perspective, without overreacting [26,27], helps to reduce feelings of distress [29,30]. This explains why self-compassion has a moderating effect on the exhaustion process.

We also found that self-compassion has a mediating effect on the relationship between job resources and engagement. This is in line with similar research showing the mediating role of self-compassion on the relationship between social support and psychological wellbeing/subjective happiness [62]. Previous research has shown that social support leads to increased self-compassion [29] and self-kindness [63]. Volunteers receive this social support through training, supervision, and co-worker support. This explains why job resources increase self-compassion. The mediating influence of self-compassion on the relationship between job resources and engagement confirms previous studies demonstrating that self-compassion increases engagement [35,36]. 

Self-compassion is a complex concept that contains multiple facets and skills: recognizing suffering, understanding the universality of suffering, feeling for one’s own suffering, tolerating uncomfortable feelings, and acting or being motivated to act to alleviate suffering [27]. Since both the moderating influence of self-compassion on the exhaustion process and its mediating role in the motivational process have been demonstrated, we examined which specific facets of self-compassion contribute to these effects. In the exhaustion process, all facets of self-compassion except understanding the universality of suffering showed a moderating effect. Earlier research among health care staff into correlations between the facets of self-compassion and distress showed that understanding the universality of suffering had the lowest correlation with distress of all facets (*r* = −0.15) [27]. In the relationship between training and engagement, all facets of self-compassion showed a mediating effect. However, acting or being motivated to act to alleviate suffering, such as taking timely breaks or seeking support from colleagues, had no mediating effect on the relationship between supervision/co-worker support and engagement. It may be that the supervision and co-worker support is more focused on promoting adequate emotion regulation skills and not so much on motivating volunteers to improve self-care.

Lastly, we looked at the added value of self-compassion on compassion towards others. Self-compassion contributed to the total explained variance of compassion towards others in both the exhaustion and motivational processes. The contribution within the exhaustion process (7%) was slightly larger than the motivational process (3%). The correlation between compassion towards others and self-compassion was weak (*r* = 0.24) but significant. This is in line with previous research [47]. Gilbert (2020) discusses that compassion towards oneself and towards others are related but different concepts. Self-compassion can be fostered by compassion towards others through an awareness of how to be sensitive and empathetic and to find out what is helpful for others and for oneself [64].

It can be concluded that self-compassion appears to be a valuable addition in the JD-R model, both in the exhaustion and the motivational processes. Although the effect of self-compassion should be further examined in longitudinal and/or experimental studies, our results suggest that it can be important for (volunteering) organizations to focus on developing self-compassion skills during supervision and training. A positive mental state is important for these volunteers so they can continue to support the crisis line organizations, which have been proven to be effective in decreasing feelings of hopelessness and psychological pain [65,66] and even in preventing suicides [65]. Further longitudinal research could demonstrate the effect of a self-compassion intervention within the framework of the JD-R model.

### 4.3. Strengths and Limitations

This is one of the first studies to evaluate the JD-R model in a volunteer organization and was conducted in a relatively large sample. However, this study also has important limitations. First, the study was cross-sectional in design, which prevents drawing conclusions about the causality or temporal nature of the relationships. Second, the questionnaire was only completed by active volunteers. Volunteers who experience more distress and less engagement may be more likely to stop volunteering. If former volunteers had completed this questionnaire, the relationships in the JD-R model might be stronger. Third, we measured the organizational outcome compassion towards others with a self-report scale which is suboptimal. An alternative would be to ask service users to assess the level of compassion and quality of received help. However, out of respect of the highly valued anonymity of the callers, this was not possible. Fourth, because this study addresses a specific group of volunteers, with specific job demands and job resources, this study is not fully representative of other groups of volunteers.

## 5. Conclusions

The findings of this study suggest that the JD-R model is applicable to volunteer organizations. Partial evidence was found for the exhaustion process and full evidence was found for the motivational process. The findings also suggest that self-compassion is a relevant personal resource for volunteers, impacting the exhaustion and motivation processes as well as compassion towards others as an organizational outcome. The findings also underscore the relevance of focusing on self-compassion during supervision and training in volunteer organizations so as to develop self-compassion skills. Further longitudinal research in various types of volunteer organizations is warranted.

## Figures and Tables

**Figure 1 ijerph-18-09651-f001:**
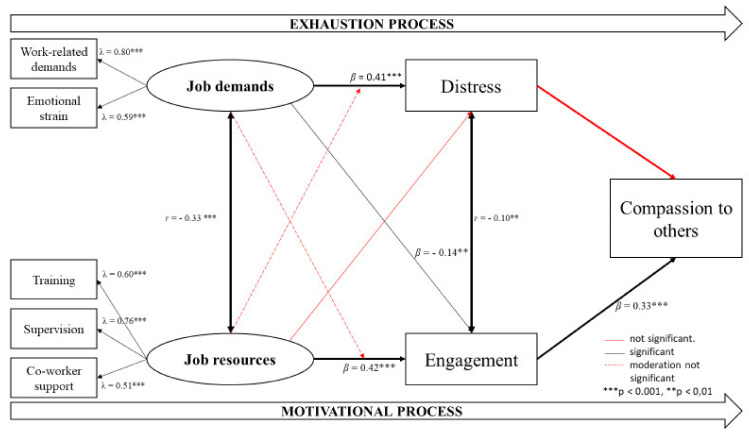
The job demands-resources model in crisis line volunteers.

**Figure 2 ijerph-18-09651-f002:**
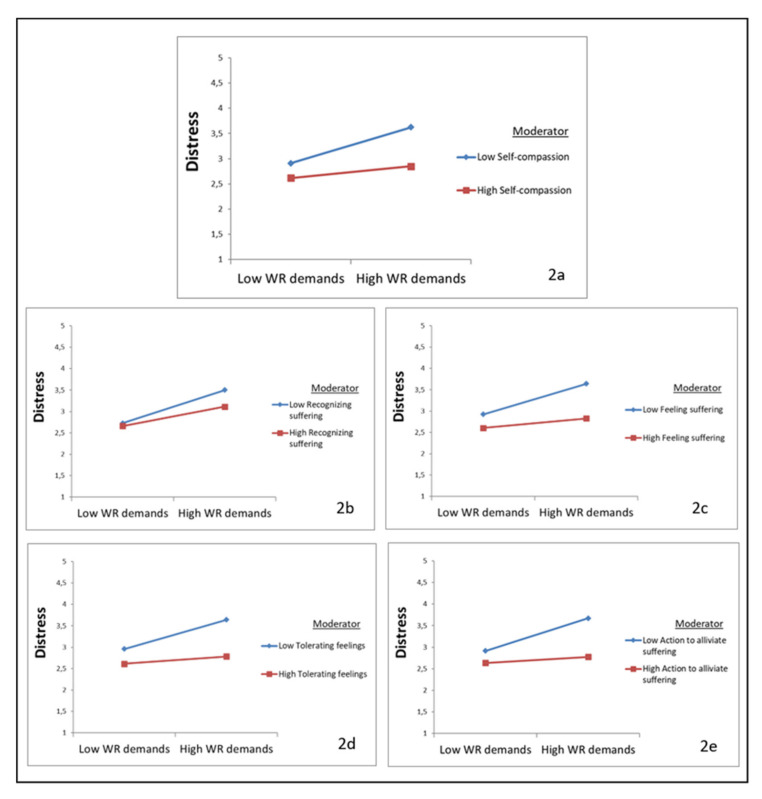
Plots of the interaction between work-related demands (WR demands) and self-compassion (and subscales) in predicting distress. (**a**) self-compassion as moderator; (**b**) Recognizing suffering as moderator; (**c**) Feeling suffering as moderator; (**d**) Tolerating feelings as moderator; (**e**) Action to alleviate suffering.

**Figure 3 ijerph-18-09651-f003:**
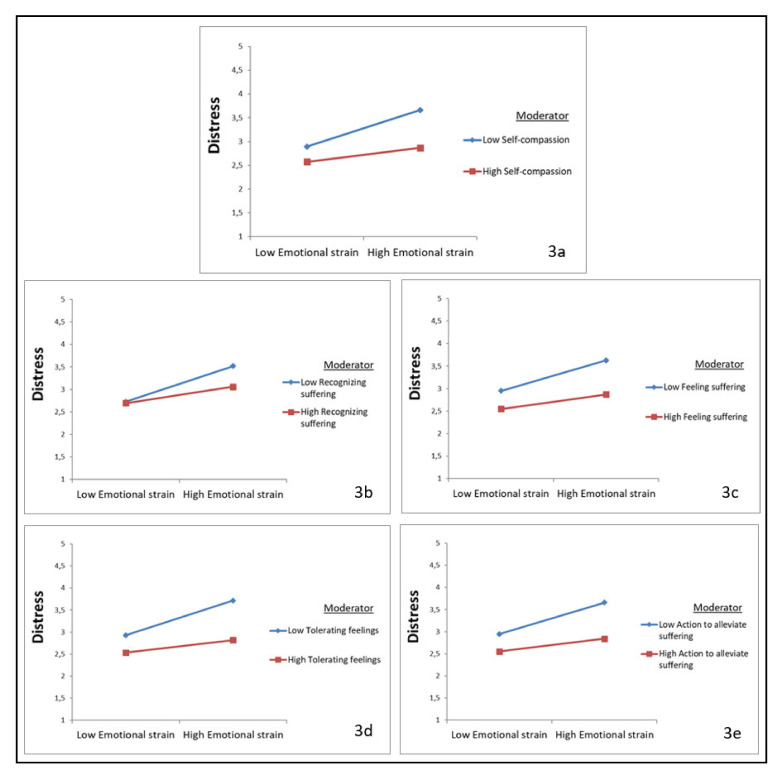
Plots of the interaction between emotional strain and self-compassion (and subscales) in predicting distress. (**a**) self-compassion as moderator; (**b**) Recognizing suffering as moderator; (**c**): Feeling suffering as moderator; (**d**) Tolerating feelings as moderator; (**e**) Action to alleviate suffering.

**Table 1 ijerph-18-09651-t001:** Demographics and work-related information (N = 543).

		Frequency	Percent
Age	18–29	10	1.8
	30–49	38	7.0
	50–64	200	36.8
	>65	294	54.1
Gender	Man	155	28.5
	Woman	387	71.3
Professional training in healthcare	Yes	196	36.1
	No	347	63.9
Experience at the crisis line	<1 year	105	19.3
	1–3 years	193	35.5
	3–6 years	89	16.4
	6–10 years	58	10.7
	>10 years	98	18.0
Hours per week	<4 h per week	97	17.9
	4–6 h per week	408	75.1
	6–8 h per week	31	5.7
	8–10 h per week	4	0.7
	>10 h per week	3	0.6
Location of work	Always on location	133	24.5
	Usually on location, occasionally at home	91	16.8
	Sometimes on location, sometimes at home	55	10.1
	Usually at home, occasionally on location	126	23.2
	Always at home	138	25.4

**Table 2 ijerph-18-09651-t002:** Descriptive statistics, correlations, and reliability coefficients for variables of present study.

			Cronbach’s Alpha	Possible Range	Mean (SD)	1a	1b	2a	2b	2c	3	4	5
Job demands	1a	Work-related demands	0.87	0–20	2.6 (1.2)								
	1b	Emotional strain	0.61	1–4	1.8 (0.3)	0.47 ***							
Job resources	2a	Training	0.88	1–5	4.2 (0.7)	−0.20 ***	−0.12 ***						
	2b	Supervision	0.82	1–5	4.3 (0.6)	−0.15 ***	−0.09 **	0.49 ***					
	2c	Co-worker support	0.73	1–4	3.7 (0.3)	−0.20 ***	−0.21 ***	0.24 ***	0.38 ***				
Strain	3	Distress	0.88	0–32	6.3 (5.5)	0.31 ***	0.28 ***	−0.11 **	−0.07	−0.17 ***			
Motivation	4	Engagement	0.91	1–7	5.0 (1.0)	−0.25 ***	−0.10 **	0.24 ***	0.34 ***	0.35 ***	−0.10 **		
Organizational outcome	5	Compassion towards others	0.76	10–100	75.2 (9.6)	−0.03	0.11 **	0.20 ***	0.27 ***	0.18 ***	0.04	0.32 ***	
Personal resource	6	Self-compassion	0.93	20–100	78.1 (9.4)	−0.17 ***	−0.10 **	0.18 ***	0.15 ***	0.17 ***	−0.31 ***	0.20 ***	0.24 ***

*** *p* < 0.001, ** *p* < 0.01.

**Table 3 ijerph-18-09651-t003:** Results of mediation models with self-compassion and subscales mediating the relationship between job resources and engagement.

Model	Job Resources	Mediator	Path A ^a^	Path B ^b^	Direct Effect ^c^	Indirect Effect ^d^	Total Effect ^e^	95% CI [LB-UB]
1	Training	Self-compassion	0.18 ***	0.16 ***	0.21 ***	0.03	0.24 **	[0.011–0.057]
		Self-compassion RS ^f^	0.11 *	0.13 **	0.23 ***	0.01	0.24 *	[0.003–0.033]
		Self-compassion UU ^g^	0.12 **	0.17 ***	0.22 ***	0.02	0.24 **	[0.004–0.047]
		Self-compassion FS ^h^	0.17 ***	0.12 **	0.22 ***	0.02	0.24 **	[0.006–0.045]
		Self-compassion TF ^i^	0.16 ***	0.15 ***	0.22 ***	0.02	0.24 **	[0.008–0.050]
		Self-compassion AA ^j^	0.15 ***	0.08	0.23 ***	0.01	0.34 **	[0.000-0.034]
2	Supervision	Self-compassion	0.15 ***	0.15 ***	0.32 ***	0.02	0.34 **	[0.007–0.047]
		Self-compassion RS ^f^	0.08	0.13 ***	0.33 ***	0.01	0.34 **	[0.001-0.028]
		Self-compassion UU ^g^	0.15 ***	0.15 ***	0.32 ***	0.02	0.34 **	[0.007–0.051]
		Self-compassion FS ^h^	0.12 **	0.12 **	0.32 ***	0.01	0.33 **	[0.002–0.037]
		Self-compassion TF ^i^	0.13 **	0.14 ***	0.32 ***	0.02	0.34 **	[0.005–0.043]
		Self-compassion AA ^j^	0.13 **	0.07	0.32 ***	0.01	-	-
3	Co-worker support	Self-compassion	0.17 ***	0.15 **	0.32 ***	0.02	0.35 **	[0.008–0.049]
		Self-compassion RS ^f^	0.10 *	0.13 **	0.34 ***	0.01	0.35 **	[0.002–0.031]
		Self-compassion UU ^g^	0.21 ***	0.13 **	0.32 ***	0.03	0.35 **	[0.009–0.056]
		Self-compassion FS ^h^	0.12 **	0.12 **	0.34 ***	0.01	0.35 **	[0.002–0.034]
		Self-compassion TF ^i^	0.13 **	0.14 *	0.33 ***	0.02	0.35 **	[0.005–0.042]
		Self-compassion AA ^j^	0.13 **	0.07	0.34 ***	0.01	-	-

^a^ Path A = regression weights relationship independent variable and mediator. ^b^ Path B = regression weights relationship mediator and engagement. ^c^ Direct effect = regression weight relationship independent variable and engagement. ^d^ Indirect effect = increase in direct effect through mediator. ^e^ Total effect = sum of direct and indirect effect. ^f^ RS = recognizing suffering, ^g^ UU = understanding universality, ^h^ FS = feeling one’s own suffering, ^i^ TF = tolerating uncomfortable feelings, ^j^ AA = being motivated to act, or action to alleviate suffering. *** *p* < 0.001, ** *p* < 0.01, * *p* < 0.05.

**Table 4 ijerph-18-09651-t004:** Summary of multiple regression analysis of the added value of self-compassion on compassion towards others.

		Exhaustion Process					Motivational Process			
	Predictor	*B*	*SE B*	*β*	*R^2^, F*	Predictor	*B*	*SE B*	*β*	*R^2^, F*
1 ^a^	Demographics and work-related variables	ns	ns	ns	R^2^ = 0.01, F (6, 536) = 0.72	Demographics and work-related variables	ns	ns	ns	R^2^ = 0.01, F(6, 536) = 0.72
2 ^b^	Emotional strain	5.61	1.66	0.17	R^2^ = 0.03, F (2, 534) = 2.01 **	Supervision	3.18	0.80	0.20	R^2^ = 0.09, F(3, 533) = 5.64 ***
3 ^c^	Work-related demands	−0.83	0.40	−0.10	R^2^ = 0.03, F (1, 533) = 1.88	Supervision	2.22	0.80	0.14	R^2^ = 0.14, F(1, 532) = 8.67 ***
	Emotional strain	5.37	1.68	0.16		Engagement	2.54	0.45	0.25	
4 ^d^	Emotional strain	5.06	1.62	0.15	R^2^ = 0.10, F (1, 532) = 5.80 ***	Supervision	2.24	0.79		R^2^ = 0.17, F(1, 531) = 9.52 ***
	Distress	0.21	0.08	0.12		Engagement	2.28	0.45	0.23	
	Self-compassion	0.28	0.04	0.28		Self-compassion	0.17	0.04	0.16	

^a^. Predictors: age, gender, professional training in health, years of experience at the CLS, hours per week working at the CLS, location of work. ^b^. Predictors: age, gender, professional training in health, years of experience at the CLS, hours per week working at the CLS, location of work, job demands (work-related demands and emotional strain) (only at exhaustion process), job resources (training, supervision, and co-worker support) (only at motivational process). ^c^. Predictors: age, gender, professional training in health, years of experience at the CLS, hours per week working at the CLS, location of work, job demands (only at exhaustion process), distress (only at exhaustion process), job resources (only at motivational process), and engagement (only at motivational process). ^d^. Predictors: age, gender, professional training in health, years of experience at the CLS, hours per week working at the CLS, location of work, job demands (only at exhaustion process), distress (only at exhaustion process), job resources (only at motivational process), and engagement (only at motivational process), self-compassion. *** *p* < 0.001, ** *p* < 0.01, ns is not significant.

## Data Availability

Data will be made available upon request.

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
