# Peer review of "The Role of Self-Compassion in the Job Demands-Resources Model, an Explorative Study among Crisis Line Volunteers"

_ijerph, 2021, doi:10.3390/ijerph18189651_

Round 1

Reviewer 1 Report

The paper deals with an interesting topic. Although the JD-R model (or theory, according to some recent literature), is well known and explored, personal resources, such as self-compassion, are less explored than organizational resources. 

The paper is well written and the structure is adequate. I have, however, some comments and suggestions that I would like to share with you, hoping that they will contribute to improve the paper.

  • please provide references for the sentence on lines 40-41
  • please complete the sentence on line 49
  • work-related demands was measured by a self-developed set of items; since it is a self-developed instrument, the authors must be more detailed in the process o creating, refining and assessing the instrument;
  • the same is valid for the measure of job resources
  • the authors present two concepts in the measurements - “engagement” and “organizational outcomes” (measured by “compassion to others”) that were not addressed in the Introduction (which in part consists of the theoretical background). This lack of theoretical background should be addressed.
  • Data analysis is quite confusing. The authors test a first model, that does not conclude self compassion, and then a second model where self-compassion is used as mediator; please be more clear in you strategy for data analysis, which implies being more clear in you goals; 
  • also, it might help to present the hypothesis of the study, since it is a quantitative research, and data analysis points to the need of hypothesis to be tested:
  • in the discussion you should avoid using new theoretical information, and new references; this problem should be fixed once you improve the theoretical background, as previous suggested;
  • the research has an important limitation that was not considered/discussed. The research focus on a very specific type of volunteer, so it is hardly a representation of voluntary work characteristics and context. 

Author Response

Dear reviewer,
Please find attached the point-by-point response to your comments and suggestions. Thank you for the valuable feedback that has led to an improvement of the article.
Kind regards,
the authors.

Reviewer 2 Report

In my opinion, this paper addresses a topic of interest and contrasts a theory, providing evidence that enriches its knowledge. I, therefore, think its publication is relevant. However, I would like to raise some doubts with the authors to clarify the work. If the authors consider that some clarification in the manuscript may be helpful, I encourage them to incorporate them. In my opinion, the paper addresses two issues to investigate.

The first is the contrast of the JD-R model to explain an outcome variable that is compassion to others. Why is it interesting to study "compassion to others" compared to other outcome variables? Unfortunately, the paper lacks a further development that justifies explaining "compassion to others" in this type of entity (Introduction section).

The second question and research is the contrast of a personal resource such as "self-compassion" to explain how it contributes to the exhaustion process (through a moderation relationship) and the motivational process (through a mediation relationship). The authors use literature on the JD-R model; however, given the type of activity, would it be possible to incorporate research evidence into a similar work environment, for example, a call centre? (For instance, see list below, Zapft et al., 1999; Bakker et al. 2003; Choi et al., 2012; Zito et al., 2018, among others).

On the other hand, the explanations proposed by the authors about the absence of effect of the health impairment process on the compassion to others variable looks pretty reasonable. One of the explanations alludes to the low level of distress, and this may have influenced. It reminds me of the arousal theory that the adverse effects of pressure take place beyond a certain threshold.

The authors highlight the weaknesses of the paper, but I believe that these do not lose the paper's interest and rigour, although they invite future research.

Minor questions:

  • Table 3. Labels RS, UU, FS, TF & AA, have not been described in the paper. Please, add the meaning of them.
  • The authors use indistinctively "compassion to others" and "compassion toward others". I suggest using only one because it is the label of the variable.

Bakker, A.B.; Demerouti, E.; Schaufeli, W.B. Dual processes at work in a call centre: An application of the job demands-resources model. Eur. J. Work. Organ. Psychol., 2003, 12 (4), 393–417.

Choi, S.; Cheong, K.J.; Feinberg, R.A. Moderating effects of supervisor support, monetary rewards, and career paths on the relationship between job burnout and turnover intentions in the context of call centres. J. Serv. Theory Pract. 2012. 22, 492–516.

Zapf, D.; Vogt, C.; Seifert, C.; Mertini,H.; Isic, A. Emotion work as a source of stress: the concept and the development of an instrument. Eur. J. Work Organ. Psychol. 1999, 8, 371–400.

Zito M.; Emanuel F.; Molino M.; Cortese C.G.; Ghislieri C.; Colombo L. Turnover intentions in a call center: The role of emotional dissonance, job resources, and job satisfaction. PLoS ONE 2018. 13 (2): e0192126. https://doi.org/10.1371/journal.pone.0192126

Author Response

(The authors gave the same response as above.)

Round 2

Reviewer 1 Report

I would like to thank the authors for their effort in addressing the suggestions and comments. The paper improved substantially. The explanation and justification of the self-developed measures could be/should be more extensive, but I don't think this is an impediment for acceptance.